# Is DEXI a Multiple Sclerosis Susceptibility Gene?

**DOI:** 10.3390/ijms26031175

**Published:** 2025-01-29

**Authors:** Anna M. Eriksson, Nora Emini, Hanne F. Harbo, Tone Berge

**Affiliations:** 1Institute of Clinical Medicine, University of Oslo, Kirkeveien 166, 0450 Oslo, Norway; annae@so-hf.no (A.M.E.); h.f.harbo@medisin.uio.no (H.F.H.); 2Department of Research, Østfold Hospital, Postboks 300, 1714 Grålum, Norway; 3Department of Research, Innovation and Education, Oslo University Hospital, Kirkeveien 166, 0450 Oslo, Norway; noraemini13@gmail.com; 4Department of Neurology, Oslo University Hospital, Kirkeveien 166, 0450 Oslo, Norway; 5Department of Mechanical, Electronic and Chemical Engineering, Oslo Metropolitan University, Pilestredet 35, 0166 Oslo, Norway

**Keywords:** autoimmunity, multiple sclerosis, susceptibility genes, *CLEC16A*, *DEXI*

## Abstract

The genetic landscape of multiple sclerosis (MS) has been extensively mapped, yielding significant insights into the molecular mechanisms of the disorder. Early studies highlighted key genes associated with the immune system, particularly T cells, as critical for MS susceptibility. Subsequent large-scale genome-wide association studies (GWASs) identified over 200 genetic variants linked to MS, revealing a complex interplay between MS risk and genes involved in various processes within adaptive and innate immune cells, as well as brain-resident microglia. Recently, a groundbreaking GWAS pinpointed the first gene variant associated with MS disease progression, distinguishing the mechanisms driving disease onset from those influencing progression. The C-type lectin domain family 16, member A (*CLEC16A*) gene within the 16p13 region has consistently been shown to be associated with increased risk of developing both MS and other autoimmune disorders. Notably, several autoimmune-associated genetic variants in *CLEC16A* introns act as expression quantitative trait loci for the dexamethasone-induced protein (*DEXI* gene, adding *DEXI* to the growing list of MS susceptibility genes. This review explores the molecular and functional characterization of DEXI with a particular focus on recent advances in understanding its role in autoimmunity, specifically in the context of multiple sclerosis. We underscore the importance of continued molecular investigation of susceptibility loci for MS identified in genetic studies, with the goal of translating this knowledge into clinical applications.

## 1. Multiple Sclerosis Genetics

The genetics of multiple sclerosis (MS) has been subjected to extensive research over the past 20 years, including large genome-wide association screenings (GWAS). These efforts have significantly enhanced our understanding of the genetic underpinnings of the disorder.

Such studies have emphasized the significance of genes with functions in the adaptive immune system, especially T cells, as associated with MS onset [1,2]. The latest MS susceptibility GWAS leveraged genotypes from 47,429 MS cases and 68,374 controls that resulted in the identification of 233 statistically independent associations with MS susceptibility that were significant genome-wide [3]. Of these variants, 32 were localized in the *HLA* gene region, and the first single nucleotide polymorphism (SNP) on chromosome X was identified. The remaining 200 genetic associations were autosomal gene variants outside the human leukocyte antigen (*HLA*) gene region, either within or near genes that are expressed in various peripheral immune cells and in resident immune cells of the brain, i.e., microglia. Using a series of methods including *cis*-expression quantitative trait locus (eQTL) analyses, 551 genes were prioritized as targeted by these 200 non-*HLA* autosomal genome-wide MS-associated variants. Biological pathway analysis of these prioritized genes identified processes beyond the previously emphasized peripheral T cell subsets, and highlighted biological processes related to development, maturation and terminal differentiation of peripheral T, B, natural killer (NK) and dendritic cells.

Until 2019, MS genetics predominantly focused on identifying susceptibility variants associated with development of the disease. As a result, we now understand 48% of its inheritability [3]. However, in 2023, a progression MS GWAS was conducted, representing a significant advancement in the field [4]. This study successfully identified the first genome-wide significant gene variant associated with MS progression. A variant in the dysferlin- zinc finger protein 638 (*DYSF-ZNF638*) locus was linked to a shortening in median time to require walking aid and increased brainstem and cortical pathology in brain tissue. Interestingly, the previously identified MS susceptibility variants were not associated with MS progression nor severity. Of note, in these genetic screenings, most of the samples were collected in Europe and the USA, and we have to acknowledge that MS genetics differ among populations. It is likely that the majority of genetic risk factors are shared, as has been seen between European/American populations and African Americans [5], but we cannot exclude the possibility that there are unidentified gene variants associated with MS susceptibility and progression that are specific for certain populations.

To sum up, genetic studies have revealed that the onset of MS is primarily driven by immunological mechanisms predominantly in the adaptive arm of the immune system, whereas disease progression appears to be influenced by processes in the central nervous system (CNS). Nevertheless, these insights set the stage for further functional studies, aiming to uncover detailed mechanistic insights into MS disease that might pave the way for developing novel prevention strategies of MS and personalized treatment based on the genetic profile of the patients.

## 2. The 16p13 Genetic Region and MS

In the latest MS susceptibility GWAS, a gene variant located at chromosome 16 (Chr16): 11213951 in the C-type lectin domain family 16, member A (*CLEC16A*) gene was identified as one of the variants most strongly associated with disease susceptibility. A second independent signal in the *CLEC16A* gene was derived from rs2286974. Additional analyses among SNPs inherited together with the identified MS susceptibility gene variants showed that the intronic *CLEC16A* rs12708716 SNP had the strongest statistical support (*p*-joined = 3.7 × 10^−71^) [3]. The *CLEC16A* gene is in the 16p13 genomic region (see Figure 1) and, since almost 20 years ago, this region has been recognized for its association with MS. Already in 2007, rs6498169 in *CLEC16A* was among the genetic variants showing suggestive association with MS in the first MS susceptibility GWAS [6]. Thereafter, *CLEC16A* has convincingly been replicated as an MS susceptibility gene in several studies [2,7,8,9,10,11,12,13,14], including the large MS GWAS published in 2011 [2] and two Immunochip studies from 2013 and 2015 [1,5], identifying SNPs, rs7200786 and rs12927355, respectively, in moderate–to-high linkage disequilibrium (LD) with rs12708716 [15].

Notably, genetic variants in *CLEC16A* have also been found to be associated with several other autoimmune disorders in addition to MS [17,18]. Therefore, many efforts have been made to understand its role in autoimmunity in general, and in MS specifically. The C-type lectin domain family 16, member A protein encoded by *CLEC16A* is instrumental for autophagic and endosomal processes whereby it affects receptor expression in NK cells and antigen-presenting cells, and T cell selection through modulation of thymic epithelial cells, impacting biological functions of importance for autoimmunity in general [19,20,21,22,23,24,25,26,27,28,29,30]. Additionally, a couple of studies analyzing loss-of-function *CLEC16A* mutations show association with neurological dysfunction due to aberrant autophagy and mitophagy [26,31,32], suggesting a role for CLEC16A also outside the immune system as thoroughly discussed in [33].

However, the MS-associated *CLEC16A* SNPs are exclusively located in introns, predominantly in intron 19, a region particularly enriched for enhancer marks in human cells [16] (see Figure 1), suggesting that this gene region may be crucial in regulating gene expression of nearby genes in addition to *CLEC16A*. The 16p13 gene region contains other genes with immune-regulatory functions. *CIITA*, which encodes the major histocompatibility complex (MHC) class II transactivator and *SOCS1* that encodes suppressor of cytokine signalling 1, both have functions of importance for autoimmunity [34,35]. The dexamethasone-induced protein (*DEXI*) gene is located between the *CLEC16A* and *CIITA* genes. Its encoded protein is far less studied than those encoded by its neighbouring genes. Interestingly, the MS-associated *CLEC16A* SNPs, or proxies thereof, was shown to act as *cis*-eQTLs for *DEXI*, and not *CLEC16A* [3]. More specifically, for SNPs in moderate-to-strong LD (*n* = 116) with the genome-wide effect SNP in *CLEC16A* (rs2286974), eight, 21 and 115 SNPs were identified as *cis*-eQTLs for *DEXI* in CD4^+^ T cells, peripheral blood mononuclear cells and the brain, respectively. Of eight SNPs with moderate-to-strong LD with the other genome-wide effect *CLEC16A* SNP (chr16: 11213951), all eight acted as *cis*-eQTLs for *DEXI* in brain tissue [3]. These findings suggest that *DEXI* may be tagged by the genetic signal within the *CLEC16A* gene, pointing to *DEXI* as a novel MS susceptibility gene. Further studies on its expression and functions are needed to understand its function in MS and autoimmunity.

## 3. Autoimmune-Associated SNPs in *CLEC16A* Act as Expression Quantitative Trait Loci for *DEXI*

Genetic mapping analyses have revealed shared associations and immune mechanisms in autoimmunity [18]. Notably, a couple of studies have identified both MS and type-1 diabetes (T1D) associated *CLEC16A* SNPs as eQTLs for *DEXI*, as summarized in Table 1. These findings suggest that *DEXI* may have a broader role in the pathogenesis of autoimmunity. The risk alleles of the MS- or T1D-associated *CLEC16A* SNPs correlated with lower *DEXI* expression in all the studies.

Additionally, we have utilized the Genotype-Tissue Expression Portal [7] to explore whether autoimmune-associated *CLEC16A* SNPs function as *DEXI* eQTLs across a diverse range of cells and tissues. We found that MS-associated SNPs, i.e., rs12708716, rs1985372, rs12927355 and rs7200786, acted as eQTLs for *DEXI* also outside brain and peripheral blood, such as in testis, breast, and adipose tissue, in arteries, and the esophagus, and that the MS risk allele correlated with reduced *DEXI* expression in these tissues as well. This indicates that autoimmune disease-associated *CLEC16A* SNPs might be in a gene region of importance for regulation of gene expression independent of cell type. Of note, intron 19 of *CLEC16A*, harbouring several of these SNPs, is highly conserved between mouse and human, which further indicates a potential regulatory function for this genetic region. Additionally, chromatin immunoprecipitation experiments show that it is bound by several transcription factors in the murine haematopoietic progenitor cell line and in human cell lines [16,36].

Interestingly, by chromatin conformation capture (3C) assays [36], Davison and colleagues showed that intronic *CLEC16A* sequences, with MS associated SNPs in intron 19, including rs12708716, are in physical proximity to the *DEXI* promoter in a monocytic cell line (THP-1 cells). High through-put chromosome conformation capture (HiC), a next generation 3-C technique, was used in lymphoblastoid cell lines (B cells) and in Jurkat T cells and showed interaction between the *DEXI* promoter and other 16p13 genetic regions harbouring autoimmune-associated susceptibility SNPs [40]. The HiC experiments showed physical proximity between the *DEXI* promoter and genetic regions harbouring the rs12928822 and rs4780401 SNPs, susceptibility variants for juvenile idiopathic arthritis and T1D, and rheumatoid arthritis, respectively. Interestingly, this study highlights that although SNPs associated with diverse autoimmune disorders are located on different chromosomal locations, they still point to the same target gene, as genetic regions far apart may be in proximity by cell-specific chromatin looping. Martin and colleagues specifically highlight *DEXI* as an example of this phenomenon [40]. As the chromatin looping experiments show the proximity between intronic *CLEC16A* sequences and the *DEXI* promoter, and with the enrichment for regulatory marks in intronic *CLEC16A* regions, it is suggested that intronic *CLEC16A* regions harbouring MS susceptibility SNPs could contain regulatory sequences such as enhancers or suppressors that may guide *DEXI* expression levels. The genotype of the MS susceptibility variants in gene regulatory regions within the *CLEC16A* gene may affect epigenetic modifications or binding of transcription factors, thereby influencing *DEXI* expression. All in all, these data suggest that *DEXI* may indeed play a role in MS and other autoimmune diseases, as is the case also for its neighbouring genes, *CLEC16A*, *CIITA* and *SOCS1*.

## 4. *DEXI* Is Ubiquitously Expressed

*DEXI* encodes dexamethasone-induced protein and was named based on the observation that its transcript was induced upon dexamethasone treatment in emphysematous tissues and in a lung tissue cell line—A549 cells—measured by semi-quantitative Northern blotting [41]. However, a couple of reports containing RNA sequencing or microarray data could not replicate this finding [42,43]. Still, its name persists.

*DEXI* was identified in 2001 as a transcript that was differentially expressed in lung tissues from patients with emphysema compared to normal lung tissues [41]. This gene seems to be ubiquitously expressed, but with higher expression in the lungs, kidney, and spleen (single cell data). When analyzing bulk RNA sequencing data, *DEXI* expression is highest in Epstein–Barr virus (EBV)-transformed B lymphocytes [7]. The latter finding aligns well with a study from 2021, showing that the EBV-encoded transcription factor Epstein-Barr virus nuclear antigen 2 (EBNA2) caused induction of *DEXI* expression in concert with a change in histone activation markers near the *DEXI* locus. The induction of *DEXI* happened in concert with reduced levels of histone activation markers at the *CIITA* locus and reduced HLA class II expression. A major EBNA2 DNA binding site was identified upstream of the *DEXI* gene and knocking out this site attenuated *CIITA* expression. As HiC data indicate that *DEXI* and *CIITA* enhancers are on different topologically associating chromosome domains, the authors suggested that enhancer competition between *DEXI* and *CIITA* represents a novel mechanism of gene regulation utilized by EBNA2 [44].

*DEXI* expression is increased in pancreatic beta cells upon the induction of inflammation and apoptosis [45] and as monocytes differentiate into macrophages [36]. We have analyzed *DEXI* expression in T cells and found that it is significantly downmodulated in peripheral CD4^+^ T cells from healthy donors and in Jurkat T cells upon activation with anti-CD3/CD28 antibody-coated beads and phorbol 12-myristate 13-acetate (PMA) and ionomycin (IO), respectively (Figure 2A,B). To establish whether this T cell activation induced reduction in *DEXI* expression was dependent of MS disease state, we analyzed *DEXI* expression in CD4^+^ T cells from both MS patients and healthy controls. CD4^+^ T cells were left untreated or activated with plate bound anti-CD3 and soluble anti-CD28 antibodies for four and 24 h. As for stimulation with antibody-coated beads (Figure 2A), this stimulation regime also resulted in a reduction in *DEXI* expression both in samples from MS patients and healthy controls (Figure 2C). However, the downmodulation of *DEXI* was slightly, but significantly, more pronounced in samples from patients with MS compared to healthy controls after four hours of activation (*p* = 0.019), despite comparable T cell activation as measured by cell surface expression of the T cell activation marker CD69 by flow cytometry. The same trend was observed after 24 h, but not significant (*p* = 0.052). Of note, *DEXI* expression was comparable in unstimulated samples between MS patients and healthy controls, in agreement with our previous findings when analyzing *DEXI* expression in freshly purified CD4^+^ and CD8^+^ T cells [46].

## 5. DEXI Function

DEXI, also known as MYLE, is a small 10.4 kDa protein consisting of 95 amino acids. *In silico* analyses indicate that DEXI comprises at least two α-helixes, with the largest of them (amino acid 25–53, see Figure 3A) identified with the highest confidence [47,48]. This central helix is predicted to be membrane-spanning, with the N-terminal part of DEXI inferred to be cytoplasmic, while the C-terminal is suggested to be non-cytoplasmic [49]. This central helix is followed by a C-terminal leucine rich sequence, which exhibits similarity to a nuclear export signal (NES) [50]. The C-terminal of DEXI may also adopt an α-helical structure, although predicted with low confidence, suggesting that some regions in its C-terminus might be unstructured [47,48].

The protein is well conserved with orthologues readily detectable in multiple species. There is 100% amino acid identity between human, rodent and bovine DEXI. Sequence alignment with DEXI protein sequences from human (*Homo sapiens*), mouse (*Mus musculus*), chicken (*Gallus gallus*) and zebra fish (*Danio rerio*) indicates that the middle sequences are particularly well conserved (amino acid 28–78) [51]. This maps to the proposed transmembrane region. Also, when searching for protein networks involving DEXI, no significant protein interactions were identified [52], making it difficult to further hypothesize about the biological function of this protein based on *in silico* findings.

Interestingly, *DEXI* knockdown reduced the incidence of apoptosis in a fibroblastic cell line treated with chemotherapy [53], but induced pancreatic beta-cell apoptosis upon viral infection [45]. Whether DEXI is involved in such a basic cellular mechanism in other models such as in immune cells remains to be studied.

Functional assessments of DEXI in relation with autoimmune disorders are limited, but there have been efforts to understand its importance for development of T1D. In one study using the nonobese diabetic (NOD) mouse model, knocking out *DEXI* had no effect on the frequency of diabetes [54]. NOD mice where *CLEC16A* was knocked down were however protected against diabetes [21]. This protection was not affected by additionally knocking out *DEXI* [54], strongly suggesting *CLEC16A* and not *DEXI* as the causal gene for the T1D associations in the 16p13 gene region. In contrast to these findings, a study presented on a preprint server for biology [55] concludes that *DEXI* disruption accelerates autoimmune diabetes in NOD mice by affecting the gut microbiome, a change that could affect the activity of immune cells involved in beta-cell destruction and susceptibility to an autoimmune attack. Of note, although they both used the NOD model, the discrepancy in their findings could be a result of differences in the animal facilities impacting the gut microbiota. Furthermore, the difference might be a result of the experimental setup. In the study demonstrating an effect of *DEXI* on autoimmune diabetes, only female mice were included in the study [55], whereas Nieves-Bonilla and colleagues did not specify the sexes of the mice used [54]. The initial stages of T1D are characterized by aberrant islet inflammation. Among the sparse available functional DEXI data, in vitro experiments show that silencing of *DEXI* in pancreatic beta cells decreases proinflammatory chemokine production, most likely via downmodulation of signal transducer and activator of transcription signalling by transcriptional activation of the interferon (*IFN*)-β promoter [45]. To conclude, additional studies are needed to clarify the function of DEXI in T1D.

Studies exploring the role of DEXI for immune cell biology in general delineate potential functional mechanisms of general importance for autoimmunity also outside T1D, for instance, in relation to MS. Since *DEXI* expression is downmodulated upon T cell activation, we hypothesized that dysregulation of *DEXI*, as impacted by disease-associated *CLEC16A* variants, could affect T cell activation. However, overexpression of *DEXI* did not affect anti-CD3/CD28, nor PMA/IO stimulated Jurkat T cell activation, as measured by cell surface expression of the T cell activation marker CD69 with flow cytometry (unpublished data from our laboratory).

Although *DEXI* mRNA expression is downmodulated upon T cell activation, the nuclear presence of the DEXI protein increases after T cell activation as can be seen from confocal microscopy images and quantified in cell fractionation experiments analyzed by Western blotting (Figure 4).

Indeed, Dos Santos et al. also showed nuclear localization of DEXI in EndoC-bH1 cells, a pancreatic beta-cell line, which was slightly increased after viral infection [45]. Also, data uploaded in the human protein atlas database show nuclear localization of DEXI, such as in U-251 MG (malignant glioblastoma cell line), A-431 (epithelial cell line) and U2OS (osteosarcoma cell line) cells [56]. The presence of a putative transmembrane helix in DEXI indicates membrane localization, whether that is the case in other cell types or under other cellular conditions remains to be studied. In conclusion, these data suggest involvement of DEXI in biological processes in the nucleus such as the regulation of gene expression. However, further analyses are required to draw definitive conclusions, as proteins can be sequestered in the nucleus and still perform their primary functions elsewhere.

## 6. Discussions and Future Studies

The MS susceptibility gene variants are associated with a low increase in MS risk (Odds ratio up to 1.3, compared to approximately 3 for major genetic risk locus *HLA-DRB15:01*). The vast majority of them are located outside exons, most likely affecting gene expression rather than gene function [3]. Although it is close to 20 years since the first of these non-*HLA* genes were identified, very few have so far been functionally deciphered and proven for relevance in MS. Three examples are worth mentioning: (i) the level of soluble tumour necrosis factor (TNF) receptor is affected by the genotype of an MS associated SNP in its gene affecting TNF-mediated disease protection [57]; (ii) an MS risk SNP near the interleukin 22 receptor alpha 2 (*IL22RA2*) gene contributes to more severe neuroinflammation in autoimmune models [58]; and (iii) an exonic tyrosine kinase 2 (*TYK2*) SNP results in reduction in pro-inflammatory signalling [59].

There is increasing evidence supporting the role of CLEC16A in various biological processes relevant to MS, such as autophagy and endosomal biogenesis, which impact T cell selection and cell surface expression of immune receptors. Although CLEC16A clearly has an important role for autoimmunity, this does not rule out the possibility that DEXI could also contribute to similar processes. Following the initial identification of *CLEC16A* as an MS susceptibility gene [6], it took nearly 15 years for functional data to be obtained, providing insights into its role in autoimmunity and elucidating the biological mechanisms through which CLEC16A exerts its effects. Only recently (in 2019) did it become clear from genetic analyses that *DEXI* is an additional MS susceptibility candidate gene in the 16p13 gene region [3]. However, as outlined above, limited functional data are available and the significance of this gene in autoimmunity remains undetermined. It may ultimately be a matter of time; an additional decade of research will allow scientists to investigate this gene within the appropriate biological context, further advancing our understanding of its role in autoimmunity.

Even though MS risk SNPs in *CLEC16A* correlates with reduced *DEXI* expression, and its T cell-induced downmodulation is affected by MS disease state, we currently lack evidence that this downmodulation has any biological impact. There are numerous unexplored biological contexts where the role of DEXI may be significant, and one such setting could be EBV infection. This infection is probably the most important environmental risk factor in MS [60]. After acute EBV infection, the virus establishes a latent infection, predominantly in memory B cells, where it persists without giving rise to disease. Notably, the cell type exhibiting the highest *DEXI* expression is EBV-transformed lymphocytes [7]. It is likely that EBV infection may induce higher *DEXI* expression in B cells, as EBNA2, an EBV-specific transcription factor, has been shown to induce *DEXI* expression at the expense of reduced *CIITA* expression, through an EBNA2-DNA recognition element just upstream of the *DEXI* promoter [44]. If this has a biological relevance is however unclear. Also, DEXI has been shown to be implicated in the regulation of local antiviral immune responses in pancreatic beta cells, specifically by participating in the transcriptional activation of *IFN-β* upon viral infection [45]. It remains to be investigated whether DEXI also plays an active role in EBV infection in B cells.

Additionally, as proteins may have cell-specific roles, DEXI may affect distinctive biological pathways in other cell types. *DEXI* has been shown to be co-expressed with its neighbouring genes, such as *CIITA*, *CLEC16A* and *SOCS1* in thymic tissues [39] and with *CLEC16A* and *SOCS1* in lymphoblastoid cells [61], indicating that these genes might act in concert at least in certain biological processes in a cell-specific manner. Whether DEXI has a function in the same pathways as the proteins encoded by its surrounding genes or in completely separate biological processes needs to be determined experimentally. Experimental autoimmune encephalitis (EAE) is the most used animal model for MS as it mirrors many of the clinical and pathological features of the disease. Induction of EAE in knock-out or humanized transgenic *DEXI* mouse models could provide valuable insights into the role of DEXI in MS disease processes. However, compensatory mechanisms can develop, making it difficult to conclude on the relevance of DEXI in this experimental setting. Therefore, such studies should be complemented by molecular studies in cellular models or patient-derived cells that could serve as platforms for studying its effect on immune cell function and response. For instance, functional genomic screens in immune cells where *DEXI* has been deleted by RNA interference or Clustered Regularly Interspaced Short Palindromic Repeats (CRISPR)-based technology can identify genes and pathways that interact with or are regulated by DEXI. As DEXI seems to be localized in the nucleus, it is possible that DEXI could act as a transcription factor modulating gene expression. Alternatively, discovering novel protein-interaction partners for DEXI could elucidate potential pathways in which DEXI operates, inferred from characteristics of the identified interacting proteins. Preliminary results from our laboratory suggest that DEXI interacts with calmodulin-modulating ligand (yeast two-hybrid screening, unpublished), suggesting a role for DEXI in calcium-dependent signalling [62]. Finally, MS is a disease affecting the CNS, and in the latest MS susceptibility GWAS, we used a large collection of dorsolateral prefrontal cortex RNA sequencing data to identify brain-specific eQTLs. Although several MS-associated *CLEC16A* SNPs and their proxies acted as immune-cell specific eQTLs for *DEXI*, an even greater fraction served as *DEXI* eQTLs when using gene expression data from the prefrontal cortex. The latest GWAS has identified MS susceptibility genes with potential roles in brain-resident immune cells such as microglia [3]. We cannot exclude the possibility that the eQTL signals from brain tissue derive from microglial *DEXI* expression. Incorporating these cells into future expression and functional studies on DEXI could be significant, as this may offer deeper insights into its potential role in MS.

## 7. Conclusions

Based on eQTL analyses, *DEXI* has been added to the growing list of MS susceptibility genes. Intronic *CLEC16A* susceptibility SNPs correlated with reduced *DEXI* expression across various cells and tissues. *DEXI* expression is also affected by external stimuli in different cells. In CD4^+^ T cells from MS patients, there is a more pronounced downmodulation of *DEXI* expression compared to healthy controls, indicating a potential dysregulation of gene expression linked to MS. It remains to be investigated whether this dysregulation is a prerequisite for the development of MS or a consequence of the disease. DEXI appears to be partly localized within the nucleus, and its presence in the T cell nucleus is increased upon T cell stimulation. Due to insufficient functional data available, it is too early to determine whether DEXI impacts critical biological pathways in MS. Understanding DEXI and the complex networks in which it functions may reveal new biological pathways relevant to MS and other autoimmune disorders, potentially guiding the scientific community toward pathways that can be targeted for novel treatment interventions.

## Figures and Tables

**Figure 1 ijms-26-01175-f001:**
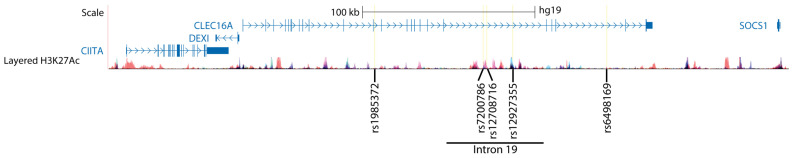
The 16p13 genomic region with the *CIITA*, *DEXI*, *CLEC16A* and *SOCS1*. The map depicts the genes in the 16p13 chromosomal region (in blue), a histone H3 histone epigenetic mark common near regulatory elements [16] (peaks in different colours), and selected MS-associated *CLEC16A* single nucleotide polymorphisms (SNPs; yellow lines). *CIITA*—major histocompatibility complex (MHC) class II transactivator, *DEXI*—dexamethasone-induced protein, *CLEC16A*—The C-type lectin domain family 16, member A, *SOCS1*—suppressor of cytokine signalling 1.

**Figure 2 ijms-26-01175-f002:**
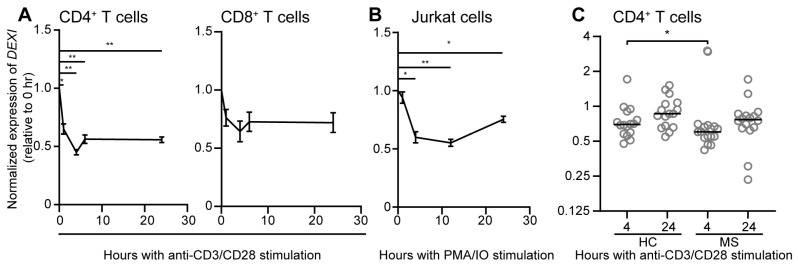
*DEXI* expression is downmodulated upon T cell activation. (**A**) CD8^+^ T cells were purified from whole blood by positive selection, followed by isolation of CD4^+^ T cells (**A**,**C**) by negative selection as described [46] and stored live on liquid nitrogen (**C**). (**A**) Freshly purified peripheral T cells were activated with anti-CD3/CD28 antibody-coated magnetic beads and (**B**) Jurkat Tag cells were stimulated with phorbol 12-myristate 13-acetate (PMA) and ionomycin (IO), for indicated time points as previously described [25], or (**C**) CD4^+^ T cells from healthy controls (HC) and MS patients [46] were stored on liquid nitrogen, thawed, and were left unstimulated or stimulated with plate-bound anti-CD3 (5 µg/mL was used for over-night coating of the wells; CD3+ Mouse-anti-human clone: OKT3, eBioscience by Thermo Fisher Scientific, San Diego, CA, USA) and soluble anti-CD28 (4 µg/mL; CD28 Purified NA/LE Mouse anti-human clone CD28.2, eBioscience by Thermo Fisher Scientific, San Diego, CA, USA) antibodies. After activation, cells were harvested, and quantitative PCR analyses performed to analyze *DEXI* expression as described in [46]. (**A**,**B**) The graphs display the mean (with standard error of the mean (SEM)) of *DEXI* expression relative to TATA-box binding protein (*TBP*) and importin-8 (*IPO*8) in CD4^+^ and CD8^+^ T cells from four healthy donors (**A**), and relative to *18S rRNA* in Jurkat cells from three independent experiments (**B**). The relative *DEXI* expression is normalized to time point 0. One-sample *t*-tests were used for comparisons. (**C**) The graph displays *DEXI* expression relative to *TBP*, normalized to the unstimulated control at four and 24 h, respectively, in samples from 16 healthy controls (HC) and 16 MS patients; the horizontal line represents the mean. Mann–Whitney tests were performed to compare *DEXI* expression between samples from HC and MS patients. * *p* <0.05; ** *p* <0.01.

**Figure 3 ijms-26-01175-f003:**
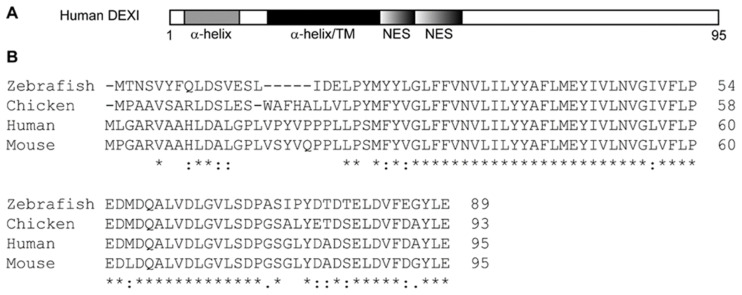
Schematic representation of the DEXI protein and sequence alignment. (**A**) The two α-helixes, predicted with high confidence, are depicted as grey and black boxes; TM = trans-membrane domain; the sequences with a potential nuclear export signal (NES) are shown as boxes with graded colour. (**B**) Protein sequence alignment of DEXI from zebrafish, chicken, human and mouse, where “*” indicates identical amino acids in all four species, “:” indicates conserved substitution of amino acids, “.” Indicates semi-conserved substitution of amino acids, and open space indicates no conservation of amino acids between the four species.

**Figure 4 ijms-26-01175-f004:**
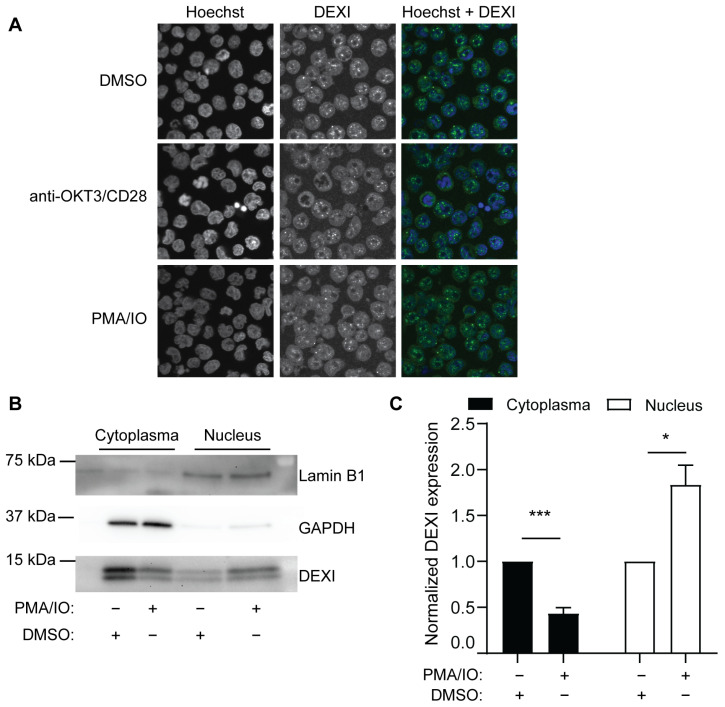
DEXI accumulates in the T cell nucleus upon activation. (**A**) Confocal images of Jurkat T cells left unstimulated (dimethyl sulfoxide, DMSO) or activated for 24 h with anti-CD3/CD28 antibodies or PMA and IO, fixed, permeabilized, and stained with antibodies against DEXI (green; Thermo Fisher Scientific, Madison, WI, USA) and the nuclear marker Hoechst (blue), as described in [25]. The images show representative cells in one out of two independent experiments. In each experiment, 500 cells were visualized. (**B**,**C**) Cytosolic and nuclear fractions were harvested according to the NE-PER Nuclear and Cytoplasmic Extraction Kit (Thermo Fisher Scientific, Madison, WI, USA) and the presence of DEXI in unstimulated and stimulated Jurkat T cells overexpressing Myc-DDK-DEXI (RC207463; Origene Technologies, Rockville, MD, USA) were analyzed by Western blotting (rabbit anti-DEXI (Thermo Fisher Scientific, Madison, WI, USA.), using antibodies against Lamin B1 (ab16048, Abcam, Cambridge, UK) and glyceraldehyde-3-phosphate dehydrogenase (GAPDH) (Santa Cruz Biotechnology, Dallas, TX, USA) as markers for the nuclear and cytosolic fractions, respectively. The signals were quantified by Image Studio Lite version 5.2.5 (Li-Cor, Lincoln, USA), and the signals from DEXI were normalized against the expression of cytosolic and nuclear markers, respectively. The graph displays the mean of the normalized signals relative to the unstimulated control (DMSO) in each fraction with SEM; * *p*-value < 0.05 and *** *p*-value < 0.001, using one-sample *t*-tests.

**Table 1 ijms-26-01175-t001:** *CLEC6A* SNPs as expression quantitative trait loci (eQTLs) for *DEXI*. The table contains SNP ID, genetic localization of the SNP within the *CLEC16A* gene, autoimmune disorders associated with the indicated SNP, the direction of the risk SNP on *DEXI* expression (*DEXI* expr.) and in which cell types the eQTLs were observed. MS = multiple sclerosis; T1D = type-1 diabetes.

SNP ID	Localization in *CLEC16A*	Disease	*DEXI* expr.	Cells	Study
rs12708716	Intron 19	T1D, MS	Lower	Brain, monocytes, B lymphoblastoid cell line	[3,36,37,38]
rs1985372	Intron 12	MS	Lower	Brain, CD4^+^ T cells	[3]
rs725613	Intron 19	T1D	Lower	Monocytes	[36]
rs3901386	Intron 17	T1D	Lower	Monocytes	[38]
rs7403919	Intron 10	T1D	Lower	B lymphoblastoid cell line	[37]
rs34306440	Intron 20	T1D	Lower	B lymphoblastoid cell line	[37]
rs6498169	Intron 22	MS	Lower	Thymic tissue	[39]

## Data Availability

The raw data are available on request from the corresponding author.

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
