# Peer review of "Is DEXI a Multiple Sclerosis Susceptibility Gene?"

_ijms, 2025, doi:10.3390/ijms26031175_

Round 1
Reviewer 1 Report
Comments and Suggestions for Authors
In this review the authors discussed molecular and functional characterization of DEXI and recent advances in understanding its role in autoimmunity. Some concerns and suggestions are listed as below:
A GWAS leveraging genotypes from 47,429 MS cases and 68,374 controls resulted in the identification of 233 statistically independent associations with MS susceptibility that were genome-wide significant. However, genetic differences were noted between Eastern and Western patients. In addition, MS is a rare disease (or may be mild) in Asian populations. This point should be discussed. Which genetic variants may play a role in clinical feature differences between Eastern and Western MS patients?
It is not clear for readers if genetic variants in CLEC16A has any relationships with clinical or radiological features of MS (for example, EDSS or white matter hyperintensities).
In the part of 'Conclusions and future studies', authors would expect more concluded statements (no need to mention T1D or mouse model in this section).
Apart from the initial identification of CLEC16A as an MS susceptibility gene, what about the role of CLEC16A in MS severity?
The authors said that limited functional data are available and the significance of this gene in autoimmunity remains undetermined. What should be done in the future? Some novel insights can be provided by authors.
The authors said that EBV infection induces higher DEXI expression, while more evidence should be provided to support this statement.
Author Response
In this review the authors discussed molecular and functional characterization of DEXI and recent advances in understanding its role in autoimmunity. Some concerns and suggestions are listed as below:
A GWAS leveraging genotypes from 47,429 MS cases and 68,374 controls resulted in the identification of 233 statistically independent associations with MS susceptibility that were genome-wide significant. However, genetic differences were noted between Eastern and Western patients. In addition, MS is a rare disease (or may be mild) in Asian populations. This point should be discussed. Which genetic variants may play a role in clinical feature differences between Eastern and Western MS patients?
We thank the reviewer for pointing out this. There are indeed genetic differences in MS disease risk between the Eastern and Western population. The MS GWAS analyses that have been discussed in our review have collected genetic information from individuals from USA, Australia, and Europe. The major genetic risk factor in MS is HLA-DRB15:01, and the frequency of HLA-DRB15:01 is 16-19% in the European and North American population, as compared to only 4-12% in Asia (for instance 4.22% among Indonesians and 11.6% among Malaysians) (Satapornpong et al., 2020, Front. Pharmacol., https://doi.org/10.3389/fphar.2020.00078). This difference, together with additional risk or protective genetic factors as well as gene-environmental interactions may be part of the explanation for the reduced incidence of MS in Asia (Zhant et al., 2023, Neurología, https://doi.org/10.1016/j.nrleng.2020.07.022). As our review focuses on one gene region identified through MS susceptibility GWASs, it is outside the scope of our article to go in depth into this topic, but we have mentioned the difference in MS genetics among populations in the revised version of the manuscript to make the reader aware of it (in section 1. Multiple sclerosis genetics, line 68-73), and we refer to other review articles for that topic. Since we focus our review on MS susceptibility and not clinical feature differences in MS; we refer also to other literature reviews for this topic.
In the revised version of the manuscript (in section 1. Multiple sclerosis genetics (line 71) and in section 2. The 16p13 genetic region and MS (line 98)), we have further chosen to refer to one of the papers trying to unravel the genetics of MS in other populations than the Western populations. In a study performed by Isobe and colleagues in African Americans, they show that the majority of MS susceptibility SNPs identified in the European/American populations acted as susceptibility SNPs also in the African American population. Of note, among others they show that rs12927355 in CLEC16A, displaying the strongest effect on MS susceptibility in the Immumnochip study on a European cohort (IMSGC, 2013, Nat. Genet., doi: 10.1038/ng.2770), displayed the most significant effect on MS in African Americans (Isobe et al., 2015, Brain, doi: 10.1093/brain/awv078).
It is not clear for readers if genetic variants in CLEC16A has any relationships with clinical or radiological features of MS (for example, EDSS or white matter hyperintensities).
To our knowledge, there are no studies that have identified CLEC16A genetic variants associated with clinical or radiological features of MS. Our review focuses on functional follow up of MS susceptibility genes identified by genome-wide association screenings. Therefore, the genetic relationships to clinical or radiological features have not been elaborated on in our manuscript. In the revised version of the manuscript, we have throughout the text done minor text corrections to make it clearer that we are focusing on MS susceptibility as well as molecular studies of DEXI. We further refer to our response about CLEC16A and severity below.
In the part of 'Conclusions and future studies', authors would expect more concluded statements (no need to mention T1D or mouse model in this section).
We thank the reviewer for pointing this out. We acknowledge that the submitted version of this section contained content that extended beyond the typical scope of a “Conclusion and future studies” section. We have now removed the T1D material and rearranged the section into two parts – Section 6: “Discussions and future studies” (line 346-429) and section 7 – “Conclusions” (line 430-443). The Conclusion section is a short section with a couple of concluding remarks. In the new Discussions and future studies section, we have pinpointed at limitations of available data regarding DEXI’s role in MS susceptibility and described how future research may improve these limitations (line 401-419), which where points raised by both reviewers. Additionally, we have included a discussion on the general challenges going from genetic identification to clinical applications, giving a couple of example genes from MS research (line 356-366).
In doing this rearrangement, we had to make smaller changes also earlier in the manuscript to make it more readable. All changes are marked in the revised version of the manuscript.
Apart from the initial identification of CLEC16A as an MS susceptibility gene, what about the role of CLEC16A in MS severity?
MS severity refers to the extent of disease symptoms from the patients, i.e. how severely a person is affected by the disease that can be measured by EDSS, disease subtypes (RRMS vs SPMS vs PPMS), lesion load on MRI etc. In the progressive GWAS (IMSGC, Nature, 2023), we also analyzed to which extent MS susceptibility variants associate with clinical parameters of relevance for severity and concluded that we do not find “evidence that susceptibility variants exert a meaningful effect on the outcome of disease”. As our review focuses on MS susceptibility and DEXI, we have not elaborated upon this, but have inserted one sentence stating the lack of association between MS susceptibility gene variants and severity (line 66-68).
Interestingly, there are functional data using lack-of function CLEC16A mutations, showing that CLEC16A seems to have a role also in neurodegeneration. This defect seems to depend on the role of CLEC16A in mitophagy, autophagy and endosomal processes in the CNS (Redmann et al., 2016, Hain et al., Sci Rep., 2021; Smits et al., Hum Genet., 2023). These data do indeed show a role for CLEC16A in the CNS, and we cannot exclude that CLEC16A has a role also in MS severity. As our review focuses on DEXI, we have not extensively elaborated upon the molecular function of CLEC16A, but we have inserted a sentence on the role of CLEC16A in neurodegeneration in the revised version of the manuscript (line 114-118). However, we refer to the original literature (Redmann et al., 2016, Hain et al., Sci Rep., 2021; Smits et al., Hum Genet., 2023) and an extensive review on CLEC16A function that was published two years ago in Internation Journal of Molecular Sciences (Pandey et al., Int J Mol Sci., 2023). The latter review is now also referred to in the revised version of the manuscript (line 118).
We would also like to emphasize, that genetics of disease progression, i.e. referring to the course of the disease over time, suggests other biological mechanisms than disease susceptibility. The genetics of MS progression vs MS susceptibility has been expanded upon in the revised version of the manuscript based on other reviewer comments (line 62-68). In the progressive MS GWAS (IMSGC, 2023, Nature), no genetic variants in the CLEC16A were identified. However, since our review focused on DEXI, we have not gone into that discussion, but we have inserted information on the role for CLEC16A in CNS related processes (line 114-118), and we already discuss the relevance of studying DEXI in CNS based on the eQTL analyses from the latest GWAS (IMSGC, Nature, 2023) (line 420-429).
The authors said that limited functional data are available and the significance of this gene in autoimmunity remains undetermined. What should be done in the future? Some novel insights can be provided by authors.
We thank the reviewer for the input and have included selected novel aspects for future research into section 6- “Discussion and future studies” in the revised version of the manuscript (line 410-419).
The authors said that EBV infection induces higher DEXI expression, while more evidence should be provided to support this statement.
We agree with the reviewer that more evidence is needed to provide a firm conclusion for EBV induced DEXI expression. In our manuscript, we have made a summary of available literature, including the paper on DEXI expression in relation to EBV infection. This is presented in section 4 and is based on one publication by Su et al in PLOS Pathogens in 2021. In short, they show that EBV infection suppresses immune recognition in naïve B-cells by down-regulating HLA genes using data from B cells from two healthy donors. They evaluate the potential mechanisms for this using among others publicly available data and CRISPR technology in EBV transformed B cells to remove a binding site for the EBV-specific transcription factor EBNA2, which is located close to the DEXI promoter. Their data supports binding of EBNA2 to this locus, resulting in down-regulation of HLA class II transcription by downmodulating CIITA gene expression. Their data further suggests that this happens through enhancer competition between topologically associated domains near the CIITA and DEXI genes, leading to enhanced DEXI expression at the expense of reduced CIITA expression. A formal proof for directly showing that after EBV infection, the EBNA2 transcription factor binds upstream of DEXI leading to increased DEXI expression and reduced CIITA expression has not been executed, and we therefore agree with the reviewer that this kind of evidence is needed to provide a firm conclusion. We have slightly modified our text in the discussions about these studies (Section 6) in the revised version of the manuscript, stating that EBV infection may induce higher DEXI expression, in addition to add some minor textual changes, labelled as track changes in the revised version of the manuscript (line 389-393).

Reviewer 2 Report
Comments and Suggestions for Authors
Dear Author, I give you the following comment. Please address this in your manuscript to enhance the readability and understanding of your manuscript.
Major Comments:
- How does the review address the potential mechanisms through which DEXI contributes to multiple sclerosis (MS) susceptibility? Is there enough evidence to support its inclusion as a definitive MS susceptibility gene?
- The review mentions CLEC16A and its relationship with DEXI. Could you elaborate on how these two genes interact and whether there is a clear, well-supported pathway linking them to MS?
- What are the limitations of the current studies cited in the review regarding DEXI’s role in MS susceptibility, and how can future research improve upon these limitations?
- The review highlights the importance of genetic studies in understanding MS. Does the review adequately discuss the challenges of translating these genetic findings into clinical applications for MS treatment?
- Could the review provide more insight into the functional consequences of DEXI variants in MS progression? How do these findings compare with other known MS susceptibility genes?
Minor Comments:
- In the abstract, the connection between DEXI and autoimmune disorders is mentioned, but the review could benefit from a clearer definition of autoimmunity in the context of DEXI’s role.
- The language used to describe the genetic associations in the review is highly technical. Could certain sections be simplified for broader accessibility?
- Are there specific studies that could be referenced to strengthen the evidence supporting DEXI as a susceptibility gene, particularly those that investigate its expression in microglia or T cells?
- The term "groundbreaking GWAS" is used in the abstract. Could this claim be further substantiated with specific details or examples of these GWAS findings?
- In the discussion of expression quantitative trait loci (eQTLs) for DEXI, could the review provide more details on the mechanisms by which these variants influence gene expression?
These questions aim to address both overarching concerns and specific technical details that could impact the robustness and clarity of the study's findings.
Best Regards
Comments on the Quality of English LanguageFine
Author Response
Dear Author, I give you the following comment. Please address this in your manuscript to enhance the readability and understanding of your manuscript.
Major Comments:
- How does the review address the potential mechanisms through which DEXI contributes to multiple sclerosis (MS) susceptibility? Is there enough evidence to support its inclusion as a definitive MS susceptibility gene?
This review focuses on the identification of DEXI as an MS susceptibility gene, concentrating on the available eQTL analyses, showing correlation between MS susceptibility SNPs in the neighboring CLEC16A gene and reduced DEXI expression. Thereafter, the review sums up available molecular and functional data on DEXI. Studies exploring the role of DEXI in immune cells would delineate potential functional mechanisms of general importance for autoimmunity also for MS. However, there are limited data on this subject, and in the revised version of the manuscript this is further emphasized. We have included a section where we elaborate on how additional studies should be performed to address the potential mechanisms through which DEXI contributes to MS susceptibility. This is written in Section 6 – “Discussion and future studies” in the revised version of the manuscript (line 401-419). Furthermore, we have, in Section 7 – “Conclusions” in the revised version of the manuscript, clearly stated that we do have genetic data indicating that DEXI is an MS susceptibility gene, but not sufficient functional data showing that dysregulation of its expression has an impact on biological pathways of importance in MS (line 430-443). Additionally, we have added a general discussion (Section 6, 356-366) on the existing challenges in going from identification to functional characterization of MS susceptibility genes.
- The review mentions CLEC16A and its relationship with DEXI. Could you elaborate on how these two genes interact and whether there is a clear, well-supported pathway linking them to MS?
As already described in the submitted manuscript, CLEC16A and DEXI are neighboring genes on chromosome 16 that are co-expressed in thymic tissue and lymphoblastoid cells. They are genetically linked to MS as CLEC16A risk SNPs correlated with reduced DEXI expression. We also describe experiments indicating no collaboration in a type-1 diabetes mouse model. Altogether, there are no functional data available linking CLEC16A and DEXI to the same biological pathways in MS. To further highlight this we included a sentence in the revised version of the manuscript (line 401-403): “Whether DEXI has a function in the same pathways as its surrounding genes or in completely other biological processes needs to be determined experimentally.”
- What are the limitations of the current studies cited in the review regarding DEXI’s role in MS susceptibility, and how can future research improve upon these limitations?
The available data show a correlation between MS risk genotypes in CLEC16A and down-modulation of DEXI expression. The main limitations of available studies are that the biological impact of reduced DEXI expression is unknown, and that we do not know the function of its encoded protein. In the revised version of the manuscript, we have based on comments from reviewer 1 changed the structure of the last section – “6 - Conclusions and future studies”. We have divided the section into two sections – “6 Discussions and future studies” and “7 Conclusions”. In Section 6 of the revised version of the manuscript, we have described the main limitations of the current studies regarding the role of DEXI in MS susceptibility and pinpointed to future studies that may improve these limitations.
- The review highlights the importance of genetic studies in understanding MS. Does the review adequately discuss the challenges of translating these genetic findings into clinical applications for MS treatment?
We thank the reviewer for pointing this out. In the revised version of the manuscript (in section 6 – Discussions and future studies, line 356-366), we have now elaborated on the challenges in the MS research field going from identification of genetic risk factors to decipher their role in the context of MS development. Here we highlight the small contribution to increased risk conferred by each gene variant, the localization of the susceptibility variants and presented the very few genes where functional data provide valuable insights into MS that may have clinical applications for MS treatment.
- Could the review provide more insight into the functional consequences of DEXI variants in MS progression? How do these findings compare with other known MS susceptibility genes?
Smaller, underpowered, studies analyzing MS susceptibility gene variants and disease severity and progression have been conducted (summarized in Waubant et al., 2019, Ann. Clin. Neurol., doi: 10.1002/acn3.50862). However, none of these studies identified DEXI as a risk factor for progression. In 2023, we published the first MS progression GWAS (IMSGC, Nature, 2023, doi: 10.1038), and none of the MS susceptibility SNPs appeared as genome-wide significant (or close to genome-wide) in the progressive GWAS. As response to reviewer 1, the findings from the progressive GWAS have been elaborated on in the introduction – section 1. “Multiple sclerosis genetics”.
Minor Comments:
- In the abstract, the connection between DEXI and autoimmune disorders is mentioned, but the review could benefit from a clearer definition of autoimmunity in the context of DEXI’s role.
We thank the reviewer for pointing this out. We have now revised this sentence in the abstract to mirror the content of the review paper (line 27-29): “This review explores the molecular and functional characterization of DEXI with a particular focus on recent advances in understanding its role as a susceptibility gene in autoimmunity, specifically in the context of multiple sclerosis.” We believe that defining the term “autoimmunity” is redundant for readers of this journal. Instead, we have emphasized “multiple sclerosis” to clarify the focus of this review, enabling the readers to specifically understand what we present in the paper.
- The language used to describe the genetic associations in the review is highly technical. Could certain sections be simplified for broader accessibility?
We partly agree with the reviewer on this point. We made these sections highly technical to avoid losing important details when describing the identification of the MS susceptibility genes in this genetic region. As CLEC16A had been the focus gene in this gene region the last 15 years, we wanted to present all the genetic details to convince the reader about the potential relevance for DEXI in MS susceptibility. In the revised version of the manuscript, we have simplified the text substantially in the following section: “2. The 16p13 genetic region and MS” and removed selected p-values and details on LD. We have chosen to keep some level of details (for instance the SNP IDs) for the readers with special interest in genetics. Overall, the sections are shortened with fewer details, and we hope this ensures broader accessibility.
- Are there specific studies that could be referenced to strengthen the evidence supporting DEXI as a susceptibility gene, particularly those that investigate its expression in microglia or T cells?
Given that MS susceptibility SNPs in CLEC16A and its proxies also function as eQTLs for DEXI in brain tissue, we cannot rule out the possibility that this signal originates from the brain-resident microglia cells. This cell type is among the expanding array of immune cells implicated in MS susceptibility, as highlighted in the MS susceptibility GWAS from 2019. Based on this, we believe it could be relevant to analyze DEXI expression and function in microglia and have now included a description about this in section 6 – “Discussion and future studies” (line 426-427) in the revised version of the manuscript.
Our data showing dysregulation of DEXI expression in CD4+ T cells from MS patients as compared to cells from healthy controls, suggest that DEXI expression might be impacted by MS, and this is now mentioned in Section 7 – “Conclusions” in the revised version of the manuscript (line 434-436), as these data support a potential role for DEXI in MS. However, whether this change in expression serves as a prerequisite for the development of MS or if it is a consequence of the disease remains to be studied and mentioned in the revised version of the manuscript (436-437).
To our knowledge there are no published studies on DEXI in microglia or T cells, and we have in the submitted manuscript provided a full overview of available data. We do however discuss the relevance of studying DEXI in microglia based on the genetic findings (line 420-425), indicating that this cell type is of particular importance for MS susceptibility and the extensive associations between CLEC16A SNPs and DEXI expression in brain derived tissues (IMSGC, 2019).
- The term "groundbreaking GWAS" is used in the abstract. Could this claim be further substantiated with specific details or examples of these GWAS findings?
We chose to use the term “groundbreaking” because the study we cite was the first GWAS that identified a genetic variant that was associated with MS disease progression, which was also mentioned in the abstract: “Recently, a groundbreaking GWAS pinpointed the first gene variant associated with MS disease progression, distinguishing the mechanisms driving disease onset from those influencing progression”. We have further substantiated these findings by providing, in brief, specific details from this GWAS in the first paragraph in section 1 – “Multiple sclerosis genetics” (line 62-68). To increase the readability of this paragraph, we had to move text (line 37-42 in the submitted version of the manuscript), to the end of the section and do some minor textual changes (all shown as “track changes”). Given that our review focuses on MS susceptibility rather than progression, we have chosen to provide more details on the genetics underlying MS susceptibility, as opposed to the findings from the progression GWAS.
- In the discussion of expression quantitative trait loci (eQTLs) for DEXI, could the review provide more details on the mechanisms by which these variants influence gene expression?
In the submitted version of the manuscript we have in section 3 described published data on chromatin conformation in the 16p13 genetic region and how chromatin looping ensures physical proximity between intronic CLE16A regions with MS susceptibility SNPs and the DEXI promoter. We have also described the potential of intronic CLEC16A sequences to act as regulatory regions based on available data from gene regulatory marks present in this region. We have in the revised version of the manuscript also added a sentence on how specifically the genotype of MS associated SNPs may affect gene expression (line 188-190): “The genotype of the MS susceptibility variants in gene regulatory regions within the CLEC16A gene may affect epigenetic modifications or binding of transcription factors, thereby influencing DEXI expression.”
These questions aim to address both overarching concerns and specific technical details that could impact the robustness and clarity of the study's findings.
We thank the reviewer for these comments and questions and have incorporated the answers into the corresponding sections in the revised version of the manuscript. We think these additions and revisions have improved the manuscript and its readability.

Round 2
Reviewer 1 Report
Comments and Suggestions for Authors
The authors have addressed my previous concerns.